# Synthesis of Valeric Acid by Selective Electrocatalytic Hydrogenation of Biomass-Derived Levulinic Acid

**Yan Du, Xiao Chen *, Ji Qi, Pan Wang and Changhai Liang ***

State Key Laboratory of Fine Chemicals, Laboratory of Advanced Materials and Catalytic Engineering, School of Chemical Engineering, Dalian University of Technology, Dalian 116024, China; sunshine233@mail.dlut.edu.cn (Y.D.); jiqi@dlut.edu.cn (J.Q.); panwang92@outlook.com (P.W.)

* Correspondence: xiaochen@dlut.edu.cn (X.C.); changhai@dlut.edu.cn (C.L.)

**Abstract:** The electrocatalytic hydrogenation (ECH) of biomass-derived levulinic acid (LA) is a promising strategy to synthetize fine chemicals under ambient conditions by replacing the thermocatalytic hydrogenation at high temperature and high pressure. Herein, various metallic electrodes were investigated in the ECH of LA in a H-type divided cell. The effects of potential, electrolyte concentration, reactant concentration, and temperature on catalytic performance and Faradaic efficiency were systematically explored. The high conversion of LA (93%) and excellent "apparent" selectivity to valeric acid (VA) (94%) with a Faradaic efficiency of 46% can be achieved over a metallic lead electrode in 0.5 M $H_2SO_4$ electrolyte containing 0.2 M LA at an applied voltage of −1.8 V (vs. Ag/AgCl) for 4 h. The combination of adsorbed LA and adsorbed hydrogen ($H_{ads}$) on the surface of the metallic lead electrode is key to the formation of VA. Interestingly, the reaction performance did not change significantly after eight cycles, while the surface of the metallic lead cathode became rough, which may expose more active sites for the ECH of LA to VA. However, there was some degree of corrosion for the metallic lead cathode in this strong acid environment. Therefore, it is necessary to improve the leaching-resistance of the cathode for the ECH of LA in future research.

**Keywords:** electrocatalytic hydrogenation; levulinic acid; valeric acid; lead

## 1. Introduction

Biomass energy, derived directly or indirectly from the photosynthesis of green plants, is a unique renewable carbon source. The development of biomass energy is of great significance for the synthesis of fine chemicals and recycling clean fuels [1–3]. The U.S. Energy Agency has selected the 12 most representative platform compounds based on more than 300 biomass-derived compounds, in which levulinic acid (LA) can be easily and economically produced from lignocellulosic materials through a simple hydrolysis process [4]. The hydrogenation of LA can (in)directly generate γ-valerolactone (GVL), valeric acid (VA), dimethyltetrahydrofuran, and 1,4-pentanediol. Among them, VA derived from the complete hydrogenation of the carbonyl group of LA can be utilized to prepare flavors, refrigerants, synthesis valerates, n-octane, valeric anhydride, and other downstream products [5]. GVL can be used in the production of lubricants, plasticizers, flavors, and solvents for insoluble resins. Therefore, the development of a transformation method of LA with high efficiency, good selectivity to high-value target products, and lower energy consumption in mild conditions is highly desirable and concurrently has an immense research worth.

Traditionally, the catalytic transformation of LA is operated through thermocatalytic hydrogenation (TCH) at high temperature and high pressure in a batch reactor using $H_2$ gas or organic solvents as the hydrogen source. Aiming at the hydrogenation of LA to VA, Li et al. reported that 97% yields of

VA and valerate esters could be achieved over a Co/HZSM-5 catalyst at 3 MPa and 240 °C for 3 h [6]. Using Ru/HZSM-5 as a heterogeneous catalyst, Weckhuysen et al. directly converted LA to VA with a yield of 45.8% in dioxane as the solvent at 200 °C for 4 h [7]. Recently, several research groups have demonstrated that the catalytic transfer hydrogenation is a promising method for the conversion of LA and its esters to GVL using several bio-alcohols or formic acid in the liquid/gas phase because it does not require the use of high pressure $H_2$ [8–11]. However, the stability of the catalyst should be considered, as it is easy to form substantial coke (catalyst deactivation) in the TCH process. In addition, the output of high energy is not conducive to sustainable development to some extent [12].

In contrast, electrocatalytic hydrogenation (ECH) operating in relatively mild conditions is a promising approach for the conversion of biomass-derived LA. Using clean and sustainable water as the hydrogen source, the reaction can be performed at atmospheric pressure and room temperature, thus avoiding the explosion of TCH and the overuse of expensive organic compounds hydrogen sources [13]. The selectivity to the target products can be controlled by adjusting the cell potential/current density, temperature, electrolyte pH, and so on [14]. Another attractive feature is that the electrical energy required for the reaction can be obtained from renewable energy, such as wind and solar [15,16]. By comparison, the thermal energy for TCH is mostly from fossil fuel resources, including coal, oil, and natural gas, which are adverse to sustainable development to a certain extent.

As can be shown from the above, the entire process of ECH is more environmental-friendly, has great research value and application prospects, and is becoming the focus of current research [12,17–20]. However, due to the competition of water splitting to generate $H_2$, the ECH of biomass conversion in aqueous solution always results in low conversion and Faradaic efficiency (FE), which means that the electron transfer is not fully used for hydrogenation of substrates but for producing $H_2$ and other byproducts. In general, the hydrogen evolution reaction (HER) mainly competes with ECH, and it is thus necessary to find a way to suppress HER or achieve high conversion and good selectivity. When it comes to the ECH of LA, Schröder first proposed a two-step electrochemical conversion of LA to octane via VA over Pb and Pt electrodes, and clarified the application of electrochemistry in the production of renewable chemicals and biofuels [21]. Subsequently, it was found that the composition of the electrolyte, the reactant concentration, and the nature of the electrode material had a strong influence on the selectivity of product formation [22]. Xin et al. conducted the ECH of LA in a single-poly-mer electrolyte membrane electrocatalytic (flow) cell reactor and found that the conversion of LA at −1.5 V vs. reversible hydrogen electrode (RHE) could reach 96.8% over a Pb electrode for 10 h through a four-electron transfer [23]. Furthermore, the addition of formic acid could increase the conversion of LA to some extent [24]. Recently, Wu reported that GVL was the only product and the FE could reach 78.6% in the ECH of LA when using PbS with different degrees of oxidation as a catalyst and 1-butyl-3-methylimidazolium tetrafluoroborate as the electrolyte [25]. However, some reaction parameters that are all important aspects of the direct conversion of LA to VA over the ECH process have not been investigated in depth. The reaction mechanism and the stability of electrodes in the acid environment also need to be examined.

Herein, the ECH of biomass-derived LA was systematically investigated in an H-type cell separated by a Nafion membrane using water as the source of hydrogen. The effects of cathode materials, potential, electrolyte concentration, reactant concentration, and temperature on the catalytic performance and FE were systematically explored. The ECH mechanism of LA to VA over metallic Pb as a cathode was confirmed. In addition, the stability of the cathode material was evaluated. This work not only contributes knowledge about the reaction system for the ECH of LA from the perspective of stability and reactivity, but also provides a meaningful guidance for the design of efficient metal electrocatalysts for large-scale biomass upgrading applications.

## 2. Results

### 2.1. ECH of LA in H-Type Cell

Generally, mutual competition and dependence relations coexist between the ECH of LA and H$_2$ evolution, which are greatly dependent on the nature of cathode materials and the choice of reduction potentials. As shown in Figure 1, various metals (Pb, Zn, Ti, Co, Pt, and Cu) as cathodes were investigated. Under the same reaction conditions, for instance, 0.2 M sulfuric acid as the electrolyte containing 0.2 M LA, −1.8 V, and reaction time 2 h, metallic Pb stands out clearly over the other materials (Zn, Ti, Co, Pt, and Cu) in terms of LA conversion to VA, with the conversion of LA reaching 56% and the selectivity to VA achieving 95% with 77% FE of VA. Although metallic Zn has the highest selectivity to VA (ca. 99%), the FE of VA (ca. 20%) is much lower than that of Pb. According to the Tafel formula ($\eta = a + b \lg i$, where $\eta$ is the overpotential (V), $i$ is the current density (A/m$^2$), and $a$ and $b$ are Tafel constants), the pure metal electrode has a fixed hydrogen evolution constant under acidic/basic conditions. It should be remarked that the best candidates for ECH may not be the metals with high overpotential for the HER, since adsorbed hydrogen is needed for the hydrogenation reaction as well. Because metallic Pt with low overpotential has an excellent ability to produce adsorbed hydrogen, it is used for the electrocatalytic reaction of phenol and furfural at a lower current potential or current density [26,27]. However, there is no denying that the ability of Pt to produce hydrogen at high potential is significantly stronger than that of ECH. Therefore, both conversion and FE are very low at −1.8 V. Surprisingly, metallic Cu with medium overpotential seems unfriendly to the ECH of LA (the conversion of LA is only 10%), even though Cu foam, a porous material with 3D structure, provides low resistance diffusion channels, promoting diffusion and ion transport of the electrolyte. In previous research, Cu-based catalysts presented outstanding performances in the ECH of furfural and 5-hydroxymethylfufural [19,28,29]. It can be seen that the affinity of different organic molecules on the same metal surface is different, which largely determines the ECH activity, rather than the overpotential for H$_2$. By comparison, the excellent ability of Pb to adsorb LA may promote the electron and electrolyte ion transfer efficiently, enhancing the activation of carbonyl in LA. Consequently, with the aim of achieving a highly efficient conversion of LA to VA through ECH over a metallic Pb electrode, the critical parameters were systematically explored.

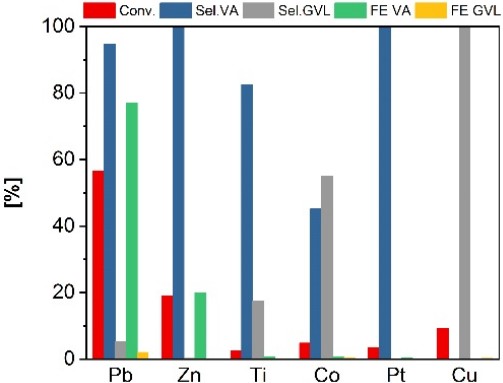

**Figure 1.** Conversion of levulinic acid (LA), selectivity to products, and Faradaic efficiency (FE) for the electrocatalytic hydrogenation (ECH) of LA to valeric acid (VA) and γ-valerolactone (GVL) at various types of metallic electrodes. Conditions: 0.2 M LA, 0.2 M H$_2$SO$_4$ electrolyte, −1.8 V vs. Ag/AgCl, 2 h.

The effect of the applied potential on the catalytic hydrogenation performance of LA was investigated at 4 h in 50 mL of 0.5 M sulfuric acid with 0.2 M LA. Figure 2 shows that the polarization extent of the metallic Pb electrode can be controlled by changing the applied potential, thus governing the products distribution, which is similar to previously reported results [23]. At −1.4 V, 84% selectivity to VA with a FE of 56% is observed for a LA conversion of 17%. At −1.6 V, the selectivity to VA increases to 92% (8% to GVL), and the LA conversion and FE to VA increase to 65% and 71%, respectively.

When the applied potential drops to −1.8 V, the selectivity to VA increases to 94%, LA conversion jumps to 93%, but FE decrases to 46%. When the potential further decreases to −2.0 V, the VA selectivity as well as LA conversion increase slightly, while the FE of VA dramatically decreases to 25%. The results demonstrate that the negative potential not only will enhance the conversion of LA but also is conducive to the formation of VA, which can be attributed to the fact that the ECH of LA involves a serial four-electron pathway through the intermediate 4-hydroxyvaleric acid [30]. The reaction intermediate can be further hydrogenated to VA or desorbed into the bulk electrolyte to form GVL. However, the FE to VA decreases significantly with the decrease in applied potential, which can be attributed to the enhancement of $H_2$ evolution at more negative potential. Therefore, the ECH of LA and the $H_2$ evolution should be balanced by tuning the applied voltage so that enough adsorbed hydrogen ($H_{ads}$) generated in situ on the electrode is present to drive the hydrogenation of carbonyl in LA.

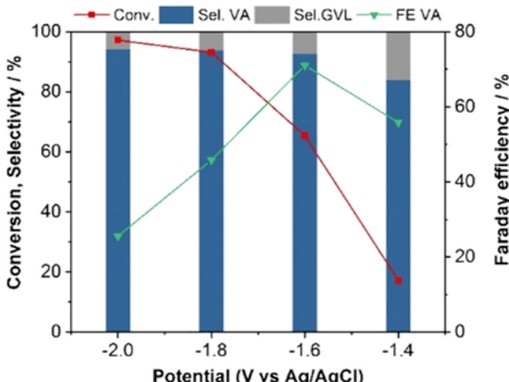

**Figure 2.** Conversion of LA, selectivity to VA and GVL, and FE (related to VA) versus different potentials. Conditions: Pb cathode, 0.2 M LA, 0.5 M $H_2SO_4$ and 4 h.

In order to understand the ECH path of LA, as shown in Figure 3, the conversion of LA as well as the yield of VA and GVL versus reaction time were examined. As the reaction time increases, the concentration of the reactant (LA) drops, and the yield of products (VA and GVL) increases. Unfortunately, carbon loss (ca. 8–18%) is observed, which may be attributed that some by-products, such as pinacol, angelica lactone, or condensate are not detected by HPLC. Furthermore, the consumption rate of the substrate gradually slows down with increasing electrolytic time, and the FE shows a downward trend as a whole, e.g., the FE of VA drops from 72% to 46% over 4 h. This can be because the concentration of LA on the surface of the working electrode decreases evidently. Under such circumstances, it is easier for the hydrion in the solution and the adsorbed hydrogen on the electrode surface to combine with each other, causing the $H_2$ evolution profits to happen.

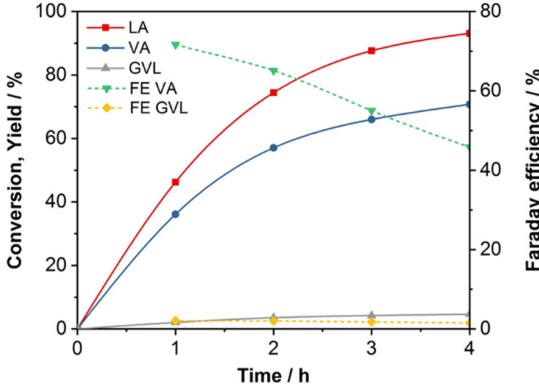

**Figure 3.** Conversion, yield, and FE of products versus the reaction time. Conditions: Pb cathode, 0.2 M LA, 0.5 M $H_2SO_4$ electrolyte, and −1.8 V vs. Ag/AgCl.

The concentration of hydrion is of vital importance in the ECH of LA. Figure 4 presents the effect of concentrations of the sulfuric acid electrolyte at the potential of −1.8 V for 1 h. The conversion of LA obviously improves with decreasing pH of the solution. When the concentration of sulfuric acid changes from 0.25 M to 1.0 M, the conversion of LA has almost doubled, which is attributed to the enhancement of electrolytic ability of water and the boost of hydrion generation. However, the FE of VA decreases slightly at lower pH. In essence, taking acid electrolyte as an example, the Volmer step (Equation (1)), in which $H^+$ gets electrons on the cathode surface to form the adsorbed hydrogen ($H_{ads}$), provides subsequent ECH steps with $H_{ads}$ as the hydrogen source. However, this $H_{ads}$ also participates in the Tafel (Equation (2)) and Heyrovsky (Equation (3)) steps.

$$\text{Volmer step}: \ H^+ + e^- \ \rightarrow \ H_{ads} \tag{1}$$

$$\text{Tafel step}: \ 2\,H_{ads} \ \rightarrow \ H_2 \tag{2}$$

$$\text{Heyrovsky step}: \ H_{ads} + H^+ + e^- \ \rightarrow \ H_2 \tag{3}$$

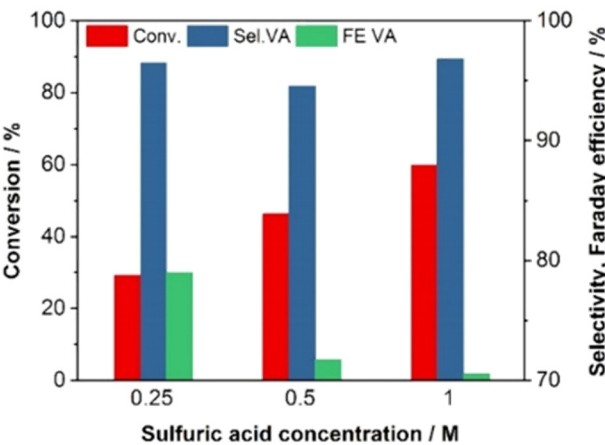

**Figure 4.** Conversion of LA, selectivity to VA, and FE of VA at different sulfuric acid concentrations as the electrolyte. Conditions: Pb cathode, 0.2 M LA, −1.8 V vs. Ag/AgCl, and 1 h.

At high hydrion concentration, more $H_{ads}$ generated on the surface of the Pb electrode has the opportunity to combine with $H^+$ to form $H_2$, which is an unwanted side reaction that reduces the density of $H_{ads}$ and consumes the applied electricity. In addition, the adsorption of LA on the catalyst site of the working electrode is also a key step for the ECH. The solution pH may affect the ionization state of LA, potentially changing its adsorption characteristics, which will influence the products distribution.

The unique catalytic ability of metallic Pb for the ECH of LA to VA was examined by comparing its linear sweep voltammograms (LSV). As shown in Figure 5a, the cathodic current density obtained without LA is due to the $H_2$ evolution. However, when LA is present, the onset potential for the reduction current density moves to the positive direction and reaches −1.22 V, suggesting that the ECH of LA is superior to the $H_2$ evolution over the metallic Pb electrode. This further confirms that the metallic Pb is a promising electrode for directional transformation of LA by electrocatalysis. Comparing the LSV curves of different starting reactant concentrations, there is no dramatic change in the current density, probably because of the limitation of electrode surface adsorption sites. However, when the starting reactant concentration is down to 0.1 M, the unwanted hydrogen evolution is more competitive with the ECH of LA, resulting in a low FE of VA (as shown in Figure 5b).

With increasing concentration of LA from 0.1 M to 0.5 M, the generated $H_{ads}$ via hydrion on the surface of the working electrode is easier to combine with organic molecules during the ECH process, leading the FE of VA to an upward trend from 51% to 72%. In addition, the selectivity to VA shows almost no change, staying at ca. 95%. However, the conversion of LA shows an obvious downward

trend from 59% to 20%. This may be due to the close contact between the active surface of the Pb electrode and the aqueous solution, which cannot provide sufficient reaction sites to withstand the higher concentration of LA hydrogenation. Overall, for sustaining the high conversion of LA and high FE of VA, the optimal LA concentration is 0.2 M over a metallic Pb electrode.

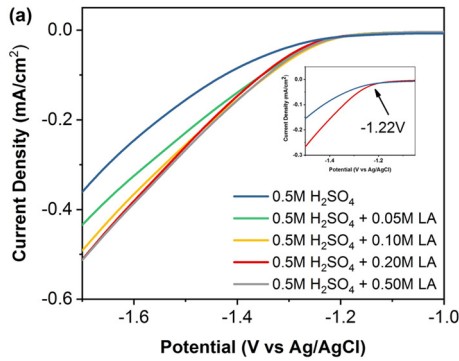 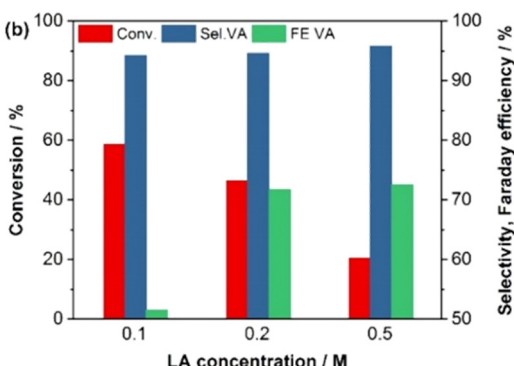

**Figure 5.** (**a**) Linear sweep voltammetry (LSV) curves with the different LA concentrations in 0.5 M $H_2SO_4$ as catholyte, and (**b**) conversion of LA, selectivity to VA, and FE of VA at different LA concentrations over Pb cathode at −1.8 V vs. Ag/AgCl for 1 h.

In the traditional TCH of LA, the equilibrium moves to the direction of the endothermic reaction when the temperature rises. The same phenomenon is found in the ECH of LA [27]. As shown in Table 1, the effect of the temperature was studied with 0.5 M sulfuric acid as catholyte at −1.8 V for 1 h. When the reaction temperature rises from 25 °C to 80 °C, the overall conversion increases from 46% to 68%, and the ECH rate shows an upward tendency (from 14 mmol $L^{-1}$ $min^{-1}$ to 23 mmol $L^{-1}$ $min^{-1}$). The high experimental temperature accelerates the movement of hydrion from the anode to the cathode, while the LA molecules involved in the reaction get more energy, which promotes the hydrogenation of LA on the surface of the metallic Pb electrode. A similar temperature effect on the conversion of feedstock and the FE of the target product has been previously reported in the ECH of furfural to furfuryl alcohol over a 3% Pt/ACF electrode using 0.1 M $H_2SO_4$ as catholyte at the potential of −0.5 V [27]. It should be noted that the $H_2$ evolution is also enhanced under the high reaction temperature (80 °C), which leads to the FE of VA (59%) being lower than that at low temperature (74% at 65 °C) due to the competition between ECH and hydrogen desorption. In addition, the selectivity to VA was also influenced by the reaction temperature due to the side reaction, e.g., the formation of GVL. Therefore, taking into account the energy consumption and the catalytic conversion efficiency, the ECH of LA to VA is preferably conducted at 65 °C over a metallic Pb cathode in 0.2 M LA in 0.5 M $H_2SO_4$ at −1.8 V for 4 h.

**Table 1.** ECH of LA with different temperatures over Pb cathode in 0.2 M LA in 0.5 M $H_2SO_4$ at −1.8 V vs. Ag/AgCl for 1 h.

| Temperature [°C] | Conversion [%] | Selectivity VA [%] | Faradaic Efficiency VA [%] | Electrocatalytic Hydrogenation Rate [mmol $L^{-1}$ $min^{-1}$] |
|---|---|---|---|---|
| 25 | 46 | 94 | 72 | 14 |
| 50 | 56 | 97 | 72 | 18 |
| 65 | 66 | 96 | 74 | 21 |
| 80 | 68 | 91 | 59 | 23 |

*2.2. Reaction Process and Mechanism*

According to the above systematic investigation and the previous literature [23], the reaction process and mechanism of ECH of LA to 4-hydroxyvaleric acid, VA, and GVL was further summarized and consummated (as shown in Figure 6). It must be emphasized that the affinity of organic molecules for the surface of the metallic Pb electrode largely determines the ECH activity [31]. In acidic electrolytes,

the adsorbed LA is activated by the $H_{ads}$ on the metallic lead surface and is then hydrogenated into the intermediate of 4-hydroxyvaleric acid. Most of 4-hydroxyvaleric acid continues to adsorb on the electrode surface to produce VA, which then desorbs from the surface of metallic Pb. A small portion of the 4-hydroxyvaleric acid intermediate desorbs from the surface of the electrode and dehydrates to generate GVL. In addition, the effect of the temperature on the ECH of LA can be well explained using this principle. Both the ECH of LA and HER are electron-getting processes as well as exothermic processes. Compared with the $H_2$ evolution, the ECH of LA to VA dominates at relatively low temperatures, which inhibits the desorption of hydrogen and enhances the combination of $H_{ads}$ with the unstable 4-hydroxyvaleric acid intermediate to form VA.

**Figure 6.** Specific reaction pathway combined with adsorption for the ECH of LA to form 4-hydroxyvaleric acid, VA, and GVL.

## 2.3. Stability Testing

Catalyst lifetime is a key consideration in the economical production of VA from bio-derived LA. Therefore, the stability of the metallic Pb electrode during the ECH of LA was further examined according to the determined reaction conditions with 0.5 M sulfuric acid containing 0.2 M LA as catholyte and 0.5 M sulfuric acid as anolyte at −1.8 V under room temperature for 4 h. After each cycle, the Pb electrode was cleaned with DI water and ethanol, and then used in the next reaction. Figure 7 reveals that there is no dramatic change in the conversion of LA (above 90%), and a very high VA selectivity of ca. 94% with a FE of ca. 48% is sustained after 8 cycles. This excellent recycling performance of the ECH of LA to VA over a metallic Pb electrode has never been reported. However, there is corrosion on the surface of the metallic Pb electrode in a strong acid environment according to a previously proposed mechanism [32,33]. The results of inductively coupled plasma atomic emission spectroscopy (ICP-AES) show that the content of Pb in the solution of the first, fourth, and eighth cycle test was 0.5032 mg/L, 5.714 mg/L, and 3.255 mg/L, respectively. The effect of metallic Pb on the structure stability was also determined using SEM images, as shown in Figure 8. The surface of fresh metallic Pb is relatively flat with traces of scratches due to the treatment with sand paper (Figure 8a,b). After stability tests, various flakes and rods are formed on the surface of the spent metallic Pb electrode (Figure 8c,d). In addition, a modification of morphology such as reconstruction may happen on the entire metal surface due to the adsorption of LA. The electrode surface becomes rough, which may expose more active sites for the ECH of LA to VA. However, the corrosion behavior of metallic Pb in the acid system would lead to the contamination of the products, creating problems for their separation and purification. It is thus necessary to design a highly efficient catalytic material with leaching-resistance for the ECH of LA to VA.

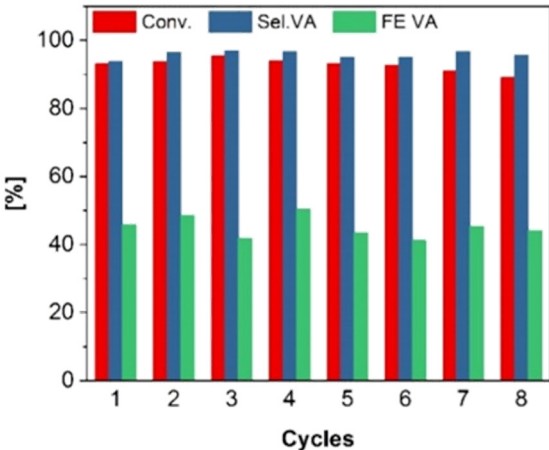

**Figure 7.** Stability testing of metallic Pb electrode in an H-type cell with 0.2 M LA as reactant and 0.5 M $H_2SO_4$ as electrolyte at −1.8 V vs. Ag/AgCl under room temperature for 4 h.

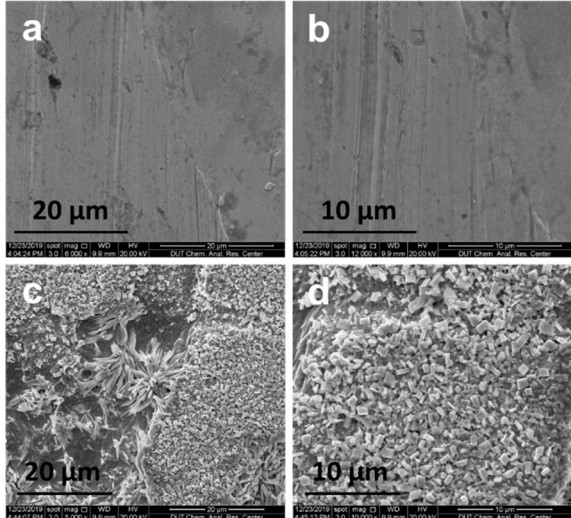

**Figure 8.** SEM images of (**a**,**b**) fresh and (**c**,**d**) spent metallic Pb electrode.

## 3. Materials and Methods

### 3.1. Chemicals and Materials

Chemicals of analytical grade were used in all experiments. LA (99.0%) was purchased from Aladdin Chemical Reagent Co., Ltd., Shanghai, China. The lead (Pb) sheet (≥99.9%) was purchased from Jinan Dingsheng Metal Material Co., Ltd., Jinan, China. Zinc (Zn), titanium (Ti), cobalt (Co) sheets (≥99.9%) were purchased from Baoji Tengfeng Metal Materials Co., Ltd., Baoji, China. The platinum (Pt) sheet (≥99.9%) was purchased from Shanghai Jingchong Electronic Technology Development Co., Ltd., Shanghai, China. Copper (Cu) foam (porosity factor 98%) was purchased from Kunshan Guangjiayuan New Material Co., Ltd., Kunshan, China. The H-type electrolytic cell was purchased from Shanghai Jingchuang Electronic Technology Development Co., Ltd., Shanghai, China. The Nafion 117 membranes used in the electrochemical device were purchased from DuPont, Wilmington, Delaware, USA. Before the electrochemical measurement, the metal sheet as an electrode was cut into a piece 1 cm × 1.2 cm, then rinsed in 3 M HCl, treated with sand paper (except Cu foam), and washed with deionized water to remove contaminants from the surface.

### 3.2. Electrochemical Measurements

All electrochemical measurements were operated on an Autolab PGSTAT302N/FRA workstation equipped with a three-electrode system. The metal sheet after pre-treatment was used as the working electrode. The platinum sheet and Ag/AgCl (0.198 V vs. standard hydrogen electrode (SHE)) were applied as counter and reference electrodes, respectively. All electrode potentials in this study were reported versus Ag/AgCl otherwise specified. Electrochemical measurements were operated in a H-type cell divided by Nafion 117 membranes with 50 mL anode and cathode chambers (Figure 9). The geometric area of the working electrode was uniformly maintained at 1 $cm^2$. During the whole experiment, the catholyte was purged nitrogen with stirring at atmospheric pressure. To investigate the ECH performance of LA over metal electrodes, linear sweep voltammetry (LSV) curves were tested with a scan rate of 50 mV $s^{-1}$ in 0.5 M $H_2SO_4$ solution with and without LA. Chronoamperometry experiments were conducted at different applied potentials and other conditions for the organic substrates' electrolysis in the reaction system.

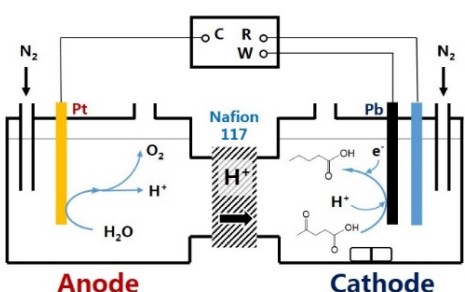 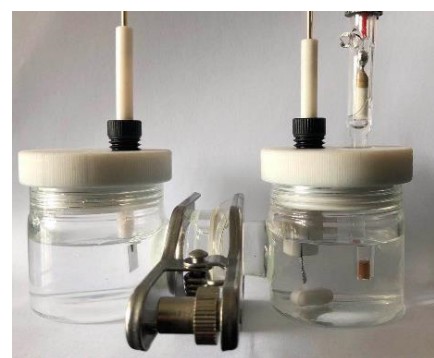

**Figure 9.** Diagram and actual photo of electrochemical device.

After the testing, the reaction products contained in the cathode electrolyte were qualitatively and quantitatively analysed by HPLC (Waters 1525) with a refractive index detector (Waters 2414). An OA-1000 column (Alltech, Grace) was operated at 85 °C, and 0.045 M $H_2SO_4$ solution was applied as eluent (0.3 mL $min^{-1}$) for product separation. A sample volume of 15 μL was injected into the HPLC system. The products were identified by comparison with standard samples. Standard curves were drawn for the quantitative analysis. The conversion of LA, the "apparent" selectivity to products, the yield of products, and FE were calculated using the following equations [23,24]:

$$\text{Conversion of LA } (\%) = \frac{\text{mol of LA consumed}}{\text{mol of initial LA}} \times 100\% \tag{4}$$

$$\text{Selectivity to X } (\%) = \frac{\text{mol of X formed}}{\text{mol of all identified products}} \times 100\% \tag{5}$$

$$\text{FE } (\%) = \frac{\text{mol of X formed}}{\text{total charge passed (C)}/(\text{F} \times \text{n})} \times 100\% \tag{6}$$

$$\text{Yield of X } (\%) = \frac{\text{mol of X formed}}{\text{mol of initial LA}} \times 100\% \tag{7}$$

$$\text{ECH rate } \left(\text{mmol L}^{-1} \text{ min}^{-1}\right) = \frac{\text{mmol of LA consumed}}{\text{volume (L)} \times \text{time (min)}} \times 100\% \tag{8}$$

where X is VA or GVL, F is the Faraday constant (96 485 C $mol^{-1}$), and *n* is the number of electrons required for the conversion of one LA molecule to one corresponding molecule of VA (*n* = 4) or GVL (*n* = 2).

*3.3. Analysis*

The morphologies of the metal electrodes before and after the reaction were characterized with a scanning electron microscopy (SEM, QUANTA 450, FEI Ltd., USA) instrument. The metal content in the reaction solution after the stability testing was measured by inductively coupled plasma atomic emission spectroscopy (ICP-AES, Optima 2000 DV device, Perkin-Elmer, USA).

## 4. Conclusions

Compared with the traditional TCH of biomass-derived LA, the ECH of LA can be achieved in relatively mild conditions with the utilization of clean and sustainable water as a hydrogen source. Over the metallic Pb electrode, the high conversion of LA (93%) and excellent "apparent" selectivity to VA (94%) with a FE of 46% can be achieved with 0.5 M $H_2SO_4$ electrolyte and 0.2 M LA at an applied voltage of $-1.8$ V (vs. Ag/AgCl) for 4 h. In order to balance the ECH of LA and the $H_2$ evolution, the adsorption hydrogen ($H_{ads}$) should be controlled precisely by tuning the cell potential/current density, temperature, and electrolyte pH, inhibiting the Tafel and Heyrovsky steps, and promoting the formation of VA. Interestingly, the reaction performance does not change significantly after eight cycles. The surface of the metallic Pb cathode becomes rough, which may expose more active sites for the ECH of LA to VA. However, the corrosion behavior of the metallic Pb electrode in the acid system leads to the contamination of the products. It is thus necessary to improve the leaching-resistance of the cathode to achieve the environmentally friendly conversion of biomass in future research.

**Author Contributions:** The individual contribution of authors to the present work was as follows: conceptualization and data analysis, Y.D. and X.C.; methodology, Y.D. and P.W.; writing—original draft preparation, Y.D.; writing—review and editing, X.C. and J.Q.; resources and supervision, C.L.; funding acquisition, X.C. and C.L. All authors have read and agreed to the published version of the manuscript.

**Funding:** This work was financially supported by the Natural Science Foundation of China (21573031 and 21703028), and the Science and Technology Innovation Fund in Dalian City (2019J12GX028).

**Conflicts of Interest:** The authors declare no conflict of interest.

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
