# Peer review of "Synthesis of Valeric Acid by Selective Electrocatalytic Hydrogenation of Biomass-Derived Levulinic Acid"

_catalysts, doi:10.3390/catal10060692_

Round 1
Reviewer 1 Report
Electrochemical conversion of levulinic acid to maximize valeric acid production was the focus of this work. Pb electrode was found to be more suitable than other metals. LA could be converted at high conversion efficiency and selectivity into VA.
Write all the chemical reactions that take place in the cell - LA to VA, GVL, 4-hydroxyvaleric acid etc. with stoichiometric balanced equations.
“However, to the best of our knowledge, few reports have been published on the direct conversion of LA to VA over the ECH process.” Didn’t you write just before that “Xin et al. conducted the ECH of LA in a PEM-based single electrocatalytic (flow) cell reactor and found that the conversion of LA at -1.5 V vs RHE could reach 96.8% over a Pb electrode for 10 h through a four-electron transfer.”?
Where is the procurement info for electrochemical reactor? Actual photos? “Autolab PGSTAT302N/FRA” This is just a potentiostat. How was the temperature controlled?
“Nafion 117 membranes” Missing in section 3.1. From where was the membrane obtained?
“purged nitrogen with stirring.” pressure?
“50 mV s−1 in H2SO4 solution” Concentration/ molarity?
“According to the Tafel formula” cite
“Cu foam with 3D porous” What does this mean?
“It can be seen that the affinity of different organic molecules on the same metal surface is different,” Where can it be seen? You don’t have results that show affinity of LA on metal surfaces.
“By comparison, the excellent adsorption of Pb to LA may promote the electron and electrolyte ion transfer efficiently” Where is the data that shows ‘excellent adsorption of Pb to LA’?
“This may be due to the close interaction between the active surface of the Pb electrode and the aqueous solution, which cannot provide sufficient reaction sites to withstand higher concentration of LA hydrogenation” What do you mean by closer interaction?
Why are your SEM images green?
SEM scale bars and the information conveyed is not visible. Point the changes with arrows.
What was the mass loss of Pb electrode after 8 cycles?
Reviewer 2 Report
The manuscript entitled “Synthesis of Valeric Acid by Selective Electrocatalytic Hydrogenation of Biomass-Derived Levulinic Acid” deals with the electrochemical reduction of levulinic acid (LA) toward the selective formation of VA. In particular, various metallic electrodes have been investigated, in this way confirming the superior activity of Pb on the target reaction. On the lead electrode a comprehensive investigation of the effects of several parameters (e.g. potential, electrolyte and reactant concentration, temperature) on the catalytic performance and Faradaic efficiency have been reported. The paper is well written and report interesting results, for these reasons I recommend this manuscript for publication in Catalysts after revisions.
More detailed comments are reported below:
- In my opinion the introduction misses to give a complete overview of alternative strategies (compare the traditional thermocatalytic hydrogenation) able to promote the reduction of levulinc acid (or its esters) to GVL or other reduced products. Therefore, I suggest completing this part adding some useful information regarding the latest results obtained in the catalytic transfer hydrogenation with bio-alcohols both in liquid and in gas phase.
- Line 110: LA is adsorbed over Pb, please rephrase.
- Figure 1: Cu electrode seems to be extremely selective toward GVL production. Please add another bar related to the FE toward GVL for all the electrode tested.
- Figure 4 shows the evolution of the relative concentration of both reagent and products by increasing the reaction time. However, from this graph the result after 2 hours seems to be quite different from the one showed in figure 1. Indeed, the LA conversion in figure 4 after 2h seems to be greater than 70%. Could the author explain this behaviour? I suggest changing this figure by plotting the variation of conversion and yield instead of the relative concentration.
- A carbon loss can be seen in figure 4 for longer reaction time (the sum of the relative concentration of the unconverted LA + VA + GVL is far from 100%). Could the author explain this behaviour? Have the authors observed the formation of 4-hydroxyvaleric acid which is claimed to be the intermediates for the formation of VA and GVL? Moreover, have the authors observed the formation of angelica lactones or of other by-products? Please add comments in the main text and (if applicable) add the evolution of by-products/intermediates in Figure 4.
- The results of the recycle tests are interesting; however, the authors should add information about the leached Pb in the reaction mixture. Please perform ICP or other suitable elemental analysis to determine the amount of Pb leached in the conditions used for the recycle experiments. Moreover, it should be of interest to evaluate if the leached lead is decreasing by increasing the catalytic cycles.
- Equation 5: selectivity is reported as the ratio between the moles of product X formed divided for the mol of “all (observed?) products”. However, in case of an uncomplete carbon balance (missing products) this selectivity is over estimated. Please change the formula by dividing the moles of product X formed by the mol of LA consumed and (if needed) change the selectivity values reported in the paper accordingly.
- Please check the manuscript for typos e.g. line 208 “taking into account”
Reviewer 3 Report
In the reviewed manuscript authors described the results of the conversion of levulinic acid to valeric acid via electrochemical reduction route. This method, due to mild process conditions, may be considered as very perspective as a one of the key element in future biorafinery strategy. Several results describing this process can already be found in the latest literature, but more in-depth analysis of several other important aspects of this process, e.g. electrode stabilising tests, have been described in this manuscript. In my opinion presented manuscript should by supplemented with:
- more detailed information on the analytical methods used in this study (post-synthesis mixtures),
- more detailed information on the reaction selectivity
Electrocatalytic hydrogenation of levulinic acid leads to many by-products. If a clear decrease in selectivity to valeric acid is observed (Table 1, Fig. 2) then what is formed?
In this context, Fig. 6 should also be corrected accordingly.
Round 2
Reviewer 1 Report
Authors have addressed all the changes satisfactorily. This manuscript can be accepted for publication.
Author Response
Dear Reviewer,
Thank you very much for your comments on our manuscript. It is my pleasure to hear from that you would recommend our manuscript to be accepted for publication.
Yours sincerely,
Xiao Chen
Reviewer 2 Report
Figures are too small, please make them bigger and with good quality.
I’m sorry but in few cases authors answer to my comments did not convince me, in particular:
Q1) A carbon loss can be seen in figure 3 for longer reaction time (the sum of the relative concentration of the unconverted LA + VA + GVL is far from 100%). Could the author explain this behaviour?
Authors Reply: Thanks for your question. The hydrogenation of LA might involve a serial four-electron pathway through the reaction intermediate 4-hydroxyvaleric acid, which has been confirm by the previous research. [1] In addition, 4-hydroxyvaleric acid easily dehydrates automatically at room temperature, so we did not observe or detect it in the HPLC. However, 4-hydroxyvaleric acid as an intermediate in the ECH of LA has been proposed by the previous reports. [2-4] For the carbon losing, it may be attribute that traces of by-products, such as pinacol, angelica lactone, or condensate are not detected by HPLC.
However, considering the “old” figure 1 I’m not convinced that the carbon balance is good enough to say that the lack is only due to “traces” of by-products. In my opinion, the authors converted the “old” figure 1 to the “new” figure 1 by changing the LA and GVL relative concentration to LA conversion and GVL yield respectively as asked. However, coming to VA yield it magically follow the trend of LA conversion with a very high selectivity, in a way that the carbon balance basically disappears (or it is very low and constant during the time). Maybe I am wrong but from the old figure one a tentative conversion of LA and VA yield can be estimated as follow:
X LA (%): 0 (0h), 46 (1h), 75 (2h), 88 (3h), 95 (4h) (as correctly reported also in the “new” figure 1)
Y VA (5): 0 (0h), 35 (1h), 56 (2h), 65 (3h), 70 (4h)
If my calculations are right, an average balance of 80% can be calculated. (calculated as the sum of the yield of detected products (GVL +VA) divided by LA conversion).
Could the authors explain these differences in VA yield compared to the ones reported in the new Figure 1? Have the author underestimated the formation of undetected by-products (up to 20% of the starting material)?
I would not take position again the way in which authors calculates the selectivity of the obtained products, however this is only an “apparent” selectivity (and this should be clearly stated in the manuscript) based on the ratio between the target product and the sum of all observed products (as stated and modified by the authors). In my opinion the authors have modified the VA yield in the new Figure 1 accordingly to their calculated apparent selectivity.
Nevertheless, authors need to clearly state the real carbon balance of their reaction system, maybe adding some data also about the VA productivity of their process.
After these modifications I think that the manuscript can be considered for publication.
